# Optical Properties of Light-Scattering Standards for CCD Photometry

**DOI:** 10.3390/s23187700

**Published:** 2023-09-06

**Authors:** Denys Bondariev, Natalia Bezugla, Paweł Komada, Nataliia Stelmakh, Mykhailo Bezuglyi

**Affiliations:** 1Department of Computer-Integrated Technologies of Device Production, National Technical University of Ukraine “Igor Sikorsky Kyiv Polytechnic Institute”, Beresteiskyi Ave., 37, 03056 Kyiv, Ukraine; denis3500722@gmail.com (D.B.); n.bezugla@kpi.ua (N.B.); n.stelmakh@kpi.ua (N.S.); m.bezuglyi@kpi.ua (M.B.); 2Department of Electronics and Information Technology, Lublin University of Technology, 38D Nadbystrzycka Street, 20-618 Lublin, Poland

**Keywords:** CCD, photometry, standards, light scattering, ellipsoidal reflector, photometric image, illuminance

## Abstract

This paper analyzes the light-scattering standards currently used for calibration (verification) and systematic research in photo and spectrophotometry tools. The application specificities in studying the diffuse reflected and transmitted light during biomedical CCD photometry are considered. The advantages of a new class of photometers with non-spherical reflectors as ellipsoids of revolution truncated along the focal planes with the internal mirror surface are presented. The ellipsoid first focal plane is combined with the surface of the under-study media, and the second is optically coupled to the CCD image sensor plane. The principles of zone analysis of spatial distribution reproduced in photometric images on a CCD sensor are substantiated. The illuminance levels of photometric image areas in reflected and transmitted light from the thickness of the standard for the wavelength of laser radiation of 650 nm of different power was experimentally investigated. Polynomial dependences were obtained, and regression coefficients of the illuminance of the external and middle rings in photometric images for the reflected and transmitted light on the laser power were determined.

## 1. Introduction

In many cases, the development of modern methods and tools for optical biomedical control focuses on the search for solutions that will improve the mechanisms of increasing the accuracy of output parameters by diagnostic equipment. In biological tissue (BT) studies, primarily non-invasive methods of light-scattering optics [1,2] have developed significantly, making it possible to analyze the optical radiation interaction with biological media and to assess their pathological conditions level. Light-scattering spectroscopy (elastic scattering spectroscopy, fluorescence spectroscopy, Raman scattering spectroscopy, multimodal spectroscopy, etc.) [3], as a prerequisite for the development of these methods, is based on a qualitative or quantitative comparison of the received light flux distributions with the known scattering and/or absorption spectra of individual elements, compounds and substances. Biomedical photometry (for example, pulse oximetry and oxygen sensors [4], bilirubinometry [5], blood glucometry [6]), in turn, uses the results of spectroscopic studies and uses one or more wavelengths in the applied quantitative analysis [7,8], which allows monitoring of the physical or metabolic indicator level. At the same time, their ability to transmittance, reflectance and spatial light scattering is considered typical biological media optical properties.

The basic approach to normalizing the illuminance level in reflected and transmitted light is verification (calibration) by measuring photometric and spectral characteristics, such as in DIN 5036-3 and CIE 130-1998, using integrating spheres. In routine practice, two-beam scanning spectrophotometers, such as Lambda 650/850/950 or Shimadzu UV-3600i Plus, are usually used to study the light scattering of solid objects. In these tools, integrating spheres are additional technical accessories that allow measuring diffuse and collimated transmission and reflection coefficients and determining absorption. The integrating spheres method allows studying of the biological tissues and scattering media optical properties and is used, as a rule, in vitro experiments [9,10,11]. Taylor’s method, based on the integrating spheres functionality, allows determining the spectral values of the diffuse reflectance coefficients in vivo [12] without the use of light-scattering standards—as the ratio of the illuminance produced during collimated irradiation of the sample to the illuminance that occurs during sphere walls irradiation. By changing the configuration of the photometer and the count of integrating spheres used, the diffuse transmission coefficient or both values are similarly determined during a single study. The diffuse reflection, total transmission and collimated transmission coefficients, determined using integrating spheres, can be a separate set of biological tissues optical properties or can be part of the input array of values used in the iterative procedures of inverse methods at the radiative transfer theory [13,14] to calculate absorption and scattering coefficients, as also the scattering anisotropy factor. Determining the scattering anisotropy factor is possible with goniophotometer measurements of the scattering phase function on thin sections of biological tissue [15,16]. At the same time, the scattering indicatrix is registered in one plane; it explains the predominant use of simple goniophotometers with the light source and the photodetector having one degree of freedom. The lack of the scattering phase function axial symmetry for some biological tissues, particularly fibrous or fibrous ones, justifies the feasibility of using integral methods for determining the scattering anisotropy factor, such as spatial photometry [17,18]. The range of values of the anisotropy factor narrows when solving the optimization task based on comparing the experimental coefficients of diffuse reflection, total transmission and collimated transmission with those calculated as modeling of light propagation in biological tissue results using the Henyey-Greenstein phase function.

As an independent method or a component of videography and aperture photometry, the CCD photometry principles are the basis for the systems designed to evaluate the brightness of point light sources or the extended zones illumination of the region of interest on a photometric image. Color or gray level, measured in pixel values, is most often associated with the object’s brightness under study [19]. In work [20], to calibrate the sensor, the traditional integrating sphere was replaced by the LRS diffuse scattering standard, and with uniform illumination by a system of xenon lamps, the illumination on the surface of the diffuser was measured using a reference photometer. Avoiding detection of specular reflection, a CCD camera with a lens and UV-IR -CUT was placed at an angle of 45° to the LRS. Cameras with CCD sensors are also used in calibration systems. For example, the authors of [21] proposed techniques, one of which is designed to measure the irradiance distribution in the focal plane, and the other which is used to determine the rays’ expansion effect passed through the transmitting diffuser.

Regardless of the type of measurement system and the experiment conditions, a set of Lambertian diffusers is most often used for calibration. For example, Calibrated Diffuse Reflectance Standards [22] contain samples with a reflectance of 98%, 70%, 50%, 20% and 2% for a wavelength of 555 nm and can be used for spectroscopic studies in the wavelength range from 250 nm to 2450 nm. It is manufactured in disks of various thicknesses with a diameter of 50.8 mm or square plates of various sizes on a metal base made of white ODM98 or pigmented from gray to black ODMP materials. Zenith Polymer diffusers [23] are manufactured of a PTFE-based material with highly Lambertian optical properties, the configuration of which allows to minimize the angular dependence and spread of the light evenly and to be used in a variety of backscattered light measurements and calibration settings in the wavelength range from 250 nm to 2500 nm. The set of measurements contains samples with diffuse reflection coefficients of 99%, 80%, 60%, 50%, 25%, 10%, 5% and 2.5%. The same manufacturer offers square Lambertian transmission sheets of different sizes with transmission coefficients of 50% (100 μm thick plate), 25% (250 μm thick plate), 16% (500 μm thick plate), 8% (1 mm thick plate) and 4% (2 mm thick plate) used in light-scattering measurement and calibration installations in the wavelength range from 250 nm to 2450 nm.

Photometers with ellipsoidal reflectors [24] are a relatively new type of measuring tool that allows the analysis of the biological tissues optical properties (scattering and absorption coefficients and scattering anisotropy factor) in reflected and/or transmitted light. The peculiarity of these photometers is the specific configuration of the collecting optical system in the form of a mirror ellipsoid of revolution with an internal reflecting surface, the first focal plane of which is aligned with the research object and the second of which is optically coupled with the CCD camera sensor. The main difference of this class of photometers compared to photometers with an integrating sphere is creating a ununiform photometric image in the CCD sensor photo-detecting plane, which characterizes the spatial distribution on the surface of the studied biological tissues in backscattered and forward-scattered light [24,25]. As practical studies have shown, the use of Calibrated Diffuse Standards gives acceptable results when determining the optical properties of biological tissues in reflected light but requires the use of ultrafine standards of scattering in transmitted light, which allow ensuring no more than 1–2 acts of scattering to determine the coefficient extinction of the biological tissues sample accurately. Given that the thickness of such ultrathin samples will have a different numerical expression for the biological tissues under study type, creating a set of such measures can cause specific difficulties. In addition, to ensure high accuracy when determining the diffuse transmission coefficient of the obtained biological tissues sample, it is advisable to choose a standard of diffuse transmission of the same thickness. Given the above, this work aims to study the optical properties of light-scattering standards made of readily available material when performing CCD photometry using a setup with ellipsoidal reflectors.

## 2. Materials and Methods

Regardless of the absolute or relative method for determining the biological tissues sample diffuse reflectance, diffuse transmission and absorption coefficients, the Taylor method and appropriate configurations of measuring devices with an integrating sphere, goniophotometer, spectrophotometer, or regular transmission and reflection measurement equipment are most often used [26]. When the coefficient of collimated transmission (reflection) is determined, there is a need to use one diaphragm/several diaphragms with small diameter openings. The size of such an opening should be smaller than or equal to (provided high accuracy of the optical system’s centering) the diameter of the incident collimated beam. These general principles are also valid when using a photometer with ellipsoidal reflectors, and the main difference is the method of analyzing the illumination formed on the sensitive plane of the photodetector.

Next, the functioning peculiarities of the photometer with ellipsoidal reflectors (ER) will be considered (Figure 1). The light collimated beam from the laser placed in the tube and the right-trapezoid prism P are directed to the biological tissues (BT) or standard. The light scattered in the backward direction by the biological tissue border located in the ER 2 first focal plane is reflected by the prism P mirror surface and forms a characteristic spatial distribution in the ER 2 s focal plane. The ellipsoidal reflector is an ellipsoid of revolution, truncated at the points of two focus orthogonally to the major axis. The hole on the side surface is made along the minor axis, and the optical center of the right-trapezoid prism coincides with the ellipsoidal reflector geometric center. The optical path within the ellipsoidal reflector internal reflective surface is the sum of the optical paths along the major and minor semi-axes of the ellipsoid. By the optical system L 2, the scattering spot is projected to the sensitive plane of the photodetector CCD 2 while the cone-shaped lens hood LH 2 performs the shielding from the background lighting function. The forward-scattered light registration is carried out similarly by a part of the photometer consisting of a reflector ER 1, a hood LH 1, a lens L 1 and a sensor CCD 1, fixed immovably on a movable base and moving in the horizontal plane to ensure the conjugation of the biological tissue another border with the ER 1 first focal plane. CCD 1 is for detecting transmission light, and CCD 2 is for reflection. The scattering spot projected on the CCD sensors is a superposition of multiple lights scattering inside the biological tissue sample and the diffuse component characterizing surface effects at the tissue boundaries.

Figure 1b illustrates the photometer with ellipsoidal reflectors experimental setup. Reflectors ER 1 and ER 2 are made by 3-D printing from ABS plastic and chemical-galvanic metallization of the internal reflective surface, the focal parameter of both reflectors is 16.7 mm, and the eccentricity is 0.7.

The conical lens hoods LH 1 and LH 2 are also printed from ABS plastic and then mechanically sanded to give the inner surface a matte finish. Serial 13VM2812ASII varifocals (Tamron Co., Ltd., Saitama, Japan) with manual aperture were used as lenses. DMK 21AU04.AS (The Imaging Source, LLC, Charlotte, NC, USA) models with an adjustable gain level with a Sony ICX098BL (Sony Corporation, Tokyo, Japan) sensor were selected as CCD cameras. Before researching, it is crucial to pre-calibrate the measuring system with ellipsoidal reflector according to the laser power level and the sensitivity of the CCD sensor, for example, using wedge photometry or Malus’s low technic. The image formation of the spatial distribution of scattered light is carried out at a zero-gain level of specialized software.

Figure 2 illustrates a typical illuminance distribution in photometric images formed in a biomedical photometer with ellipsoidal reflector. The integral characteristics of diffuse reflection and transmission (total and collimated) on the light-scattering standards obtained during reference measurements are formed by the total value of bright pixels across the photometric images entire field. They are correlated with the corresponding values of the diffuse reflectance, total and collimated transmittance coefficients. These coefficients with the albedo and scattering anisotropy factor [17] are used to determine the media’s scattering and absorption coefficients by inverse Monte Carlo simulation [25].

Figure 2 also shows the elements characterizing the principles of processing photometric images. The systematic research of standards of different thicknesses and direct measurements with samples of biological tissues involves step-by-step adjustment of the biomedical photometer information and measurement system. It leads to the need to define a region of interest and a point of radial symmetry for each series of 2D pictures, which are photometric images. The region of interest is a square inscribed with a circle with a radius equal to the ellipsoid focal parameter. The procedure for region of interest is identified as follows: First, a pixel-by-line analysis of the photometric image and determining the bright area boundaries by pixel coordinates; secondly, compliance with the form objective function, as the ratio of the square sides in which the bright area is located, acts as a condition. Suppose the value of the objective function does not meet the set limits. In that case, it is necessary to lower the sensitivity threshold, which is the value of the brightness ratio of the current and maximum bright pixels and repeat the line-by-line analysis. Manipulation of the objective form function and the sensitivity threshold allows for region of interest successfully determining for most images. When the region of interest is bounded by a square whose side contains an odd number of lines of pixels, the radial symmetry point is the photometric image’s central pixel. Otherwise, the radial symmetry point is formed by the four central pixels of the photometric image. The radial symmetry point is used when calculating the brightness level of the image’s separate areas, which have the circles shape with radii R1, R2 and R3. The radius R1, as a rule, corresponds to the incident collimated beam radius (or the circle radius in which a small rectangle is inscribed, as shown in Figure 2b to measure in the reflected light) while the illumination of circle A1 is maximum. The radii R2 and R3 for the proportionality of the illuminance level of the photometric image package are set to be the same for each image. The illuminance level is defined as the ratio of pixel values of bright points placed in rings A2 and A3, limited by a pair of radii R1 and R2, R2 and R3, to the area of these rings, respectively. The tube and the prism P falling into the lens L 2 field of view leads to darkened rectangles appearing in the photometric images on the CCD 2 (Figure 2b). Due to systematicity, it does not significantly impact the rings A2 and A3 illuminance level determination in reflected light. For the automated packeted determination of illumination in rings A2 and A3, the authors developed specialized software, similar to the IRIS of Christian Buil’s, for processing photometric images in reflected and transmitted light.

## 3. Results and Discussion

For various optical experiments, when conducting reference measurements, polymers or solutions containing light-scattering particles of various sizes in appropriate concentrations are often used, which directly affect the scattering and absorption coefficient values. Such polymers include polycaprolactone (PCL), polydimethylsiloxane (PDMS), polyethylene (PE), polyethylene terephthalate (PET), polystyrene (PS), polytetrafluoroethylene (PTFE), polyvinyl alcohol (PVA) and others as the biological tissue phantoms [27,28]. According to the structure, they can acquire a liquid or solid state. The superimposition of polymers layers allows not only the selection of the appropriate phantom thickness but also the modeling of the structure, for example, of collagen fibers, the most common protein in the human body. This study chose a 0.1 mm thickness polyethylene as a standard, and the sequential combination of polyethylene layers simulated different thickness standards. For a laser diode with a nominal power of 25 mW at a wavelength of 650 nm, an electrical system for sampling the output power with a step of 2.4 mW was developed, controlled by a reference optical power meter StarBright type. The laser diode was part of a module with a collimating lens system, providing an incident beam diameter of 2 mm. As a result of a series of experiments, photometric images were obtained in reflected and transmitted light for different thicknesses from 0.1 mm to 10 mm and at different laser radiation power (Figure 3).

As shown in Figure 3, the illuminance level of photometric images in transmitted light decreases noticeably within rings A2 and A3; conversely, in reflected light, the illuminance changes slowly. The photometric images in Figure 3a,b correspond to the thickness of the polyethylene film sample with a thickness of 2 mm, Figure 3c,d—5 mm, Figure 3e,f—8 mm and were obtained at the 4.2 mW optical power. The characteristic distribution and the outlined defining trend of illuminance are fully correlated with the model interpretation of the photometer with ellipsoidal reflectors and with its practical implementations in the study of various biological tissues [24,25]. To quantify the research results, we will analyze the illuminance level of rings A2 and A3 depending on the polyethylene investigated sample’s thickness at different incident optical radiation power. Measurement was carried out at three points along the polyethylene sample perimeter by three capture pictures, and the illuminance values were averaged. Figure 4 shows the dependence of the illuminance of rings A2 and A3 on the thickness of the examined film samples in transmitted light, and Figure 5–is reflected.

As a regression analysis result, the expressions for describing the illuminance of rings A2 and A3 in transmitted and reflected light the following equations can represent:(1)$$E_{T}(d)=me^{nd},$$
(2)$$E_{R}(d)=a\cdot ln(d)+b,$$
where $E_{T}(d)$ is the illuminance of the A2 or A3 ring for photometric images in transmitted light; $E_{R}(d)$—in reflected light; *d*—thickness of PE film sample, mm; *m*, *n*, *a* and *b* are coefficients that depend on the laser beam power.

Formula (1) characterizes the approximation for the samples with thicknesses greater than 0.7 mm.

For the graphs shown in Figure 4, coefficients *m* and *n* are polynomials by the type:(3)$$m=m_{2}P^{2}+m_{1}P+m_{0},$$
(4)$$n=n_{6}P^{6}+n_{5}P^{5}+n_{4}P^{4}+n_{3}P^{3}+n_{2}P^{2}+n_{1}P+n_{0},$$
where *P*—is the power of the laser beam, mW.

Graphs in Figure 5 are characterized by coefficients *a* and *b*, which are polynomials by the type:(5)$$a=a_{2}P^{2}+a_{1}P+a_{0},$$
(6)$$b=b_{2}P^{2}+b_{1}P+b_{0}$$

For the used light source and discrete power levels, the polynomial coefficients are presented in Table 1.

It is evident that, when restoring the illuminance distribution function from the laser power for different thicknesses media samples, the photometric image analysis consists of determining the integral gray level over the image’s entire field within the circular region of interest with a radius of R3. At the same time, wedge photometry, as a calibration prerequisite for the functioning of photometry with ellipsoidal reflectors, is aimed at obtaining the range of the linear section of the CCD light intensity response curve. This aspect is fundamental for determining the optical coefficients of the sample under study. When observing the behavior of the coefficients m, n, a and b in the equations of illuminance of the middle and external ring in transmitted and reflected light, from a physical point of view, it is possible to ascertain the nonlinear dependence of the amount of scattered light on the laser power in different zones of the photometric image. The scattering properties of the selected standards of the corresponding thicknesses can fully explain this. At the same time, the authors predict the existence of a relation between the illuminance and the concentration of scattering centers in the standards, the absolute values of which increase proportionally with the increase in the thickness of the sample.

The numerical relation between the illuminance level of different areas of images by photometry with ellipsoidal reflectors [25] and the optical properties of the studied samples, such as scattering and absorption coefficients as the scattering anisotropy factor, still needs to be clarified. The integrating capabilities for calculating the biological tissues’ optical properties using inverse Monte Carlo and photometer with ellipsoidal reflectors have made it possible to obtain reliable results for structural studies [17,24] and to register the dynamic changes in health indicators [29]. In the obtained results analysis, particularly the relation between the power of the laser beam and the illuminance distribution depending on the thickness, it is possible to note certain comparability and affinity with the transmittance and reflectance values of Ag_50_:Bi_50_ multilayer coatings [30]. Optical transmission spectra of as-deposited MoO_3_ thin films of various thicknesses (100 nm, 200 nm and 400 nm) [31] at a wavelength of 650 nm showed a similar trend to the transmission value, and the decrease in the dependence of the energy characteristics is fully explained by the increase in homogeneity and the decrease in centers scattering for thicker samples. Modeled and experimentally determined transmittance and reflectance spectra of evaporated Ge_30_Se_70_ thin films [32] do not have a pronounced dependence on thickness, which can be explained by the presence of a strong absorption region in this spectral range, which, however, is a reasonably typical feature for this films type. The extinction coefficient calculated based on transmittance measurements for Ge–Se–Te thin films [33] with a thickness of 0.8 and 1.1 mm at the used wavelength correlates with the obtained illuminance values (Figure 4).

The works [30,31,32,33] present the results of spectral measurements of films of different thicknesses under different conditions of an optical experiment, which outlines for the authors of this paper the prospects for further research on the search for spectral dependence in the illuminance distribution over the photometric image areas in both forward-scattered and backscattered light. The photometric image illuminance, considered on the example of a cheap and available PE film as light-scattering standards in transmitted and reflected light, has a characteristic distribution in separate zones of the CCD sensor detecting plane. Aberration analysis of the projecting properties of an ellipsoidal mirror with a lateral working surface can be analyzed, for example, using ray-tracing principles [34,35]. It makes it possible to assess the involvement degree of the relevant part of the ellipsoidal reflector and the peculiarities of forming the corresponding illuminance level in the sensor’s CCD field and to form dependencies set for practical use in determining the optical coefficients of biological tissues.

Spectral density values characterize the optical properties of biological media. At the same time, photometry using monochrome and color CCD cameras is implemented to analyze illumination by shades of gray. Optimizing the functionality of the photometer with ellipsoidal reflectors and ensuring acceptable accuracy of photometry comes down to the correct selection of the CCD sensor and the range of the used power and spectrum of collimated radiative sources. At the same time, the authors foresee the need to develop a base of reference measurements for different classes of light-scattering media (conditionally more scattering, less scattering and intermediate values) in the working spectral range of a photometer-specific configuration.

## 4. Conclusions

The optical measurements result in the form series of experimental photometric images with characteristic illumination distribution over the field carried out using a CCD photometer, the ellipsoidal reflectors of which were obtained by 3D printing from ABS plastic with subsequent chemical–galvanic metallization for an internal mirror surface creation. The optical properties of the scattering media were investigated using the different thicknesses of polyethylene film at the laser radiation wavelength of 650 nm in the variable power range of 1.8–23.4 mW by analyzing the illuminance level of the photometric image circular zones—in particular, the middle and external rings. The resulting distribution showed a logarithmic dependence of the illuminance of the inner and external ring in backscattered light and an exponential dependence in forward-scattered light. At the same time, the relation between the laser radiation power level and the change in illuminance has a polynomial dependence, which can be determined for a specific type of collimated flow and the light source wavelength. The work results prove the effectiveness of using a photometer with ellipsoidal reflectors as an alternative method of researching available and cheap light-scattering standards and, therefore, in determining optical coefficients in a biomedical experiment.

## Figures and Tables

**Figure 1 sensors-23-07700-f001:**
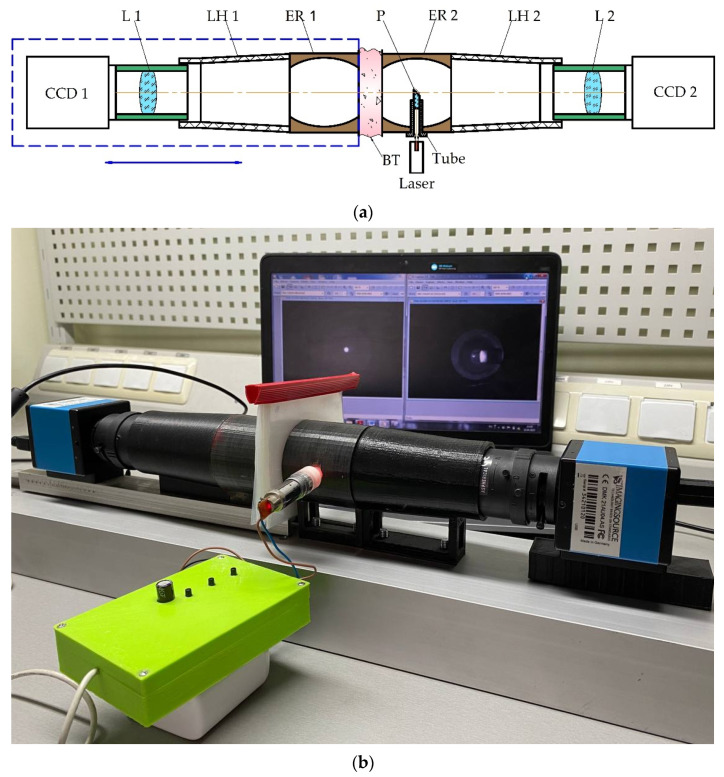
Photometer with ellipsoidal reflectors for registration of scattered light in forward and backward directions: (**a**) structural scheme; (**b**) experimental setup (L–lens, LH–hood, ER–reflector, P–prism, BT–biological tissues).

**Figure 2 sensors-23-07700-f002:**
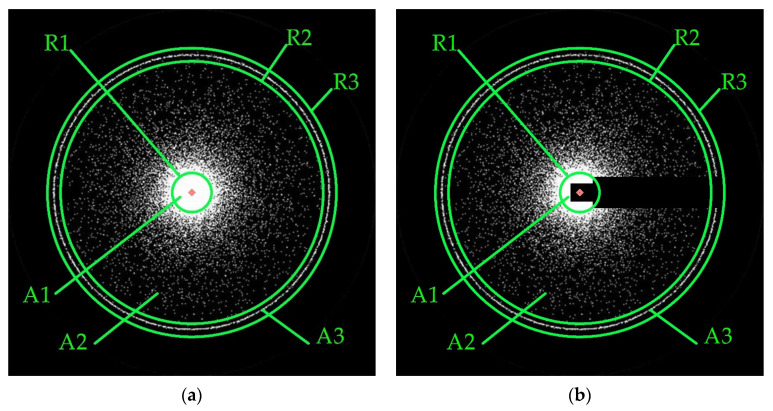
Photometric images formed on the CCD sensors of the photometer with ellipsoidal reflector: (**a**) in forward-scattered (transmitted); (**b**) backscattered (reflected) light. R1 is the radius of the collimated flow; R2 and R3 are the radii limiting the peripheral illuminance zone in the photometric image; A1 is the area of a circle with radius R1; A2 is the area of the middle ring bounded by circles R1 and R2; A3 is the area of the external ring bounded by circles R2 and R3.

**Figure 3 sensors-23-07700-f003:**
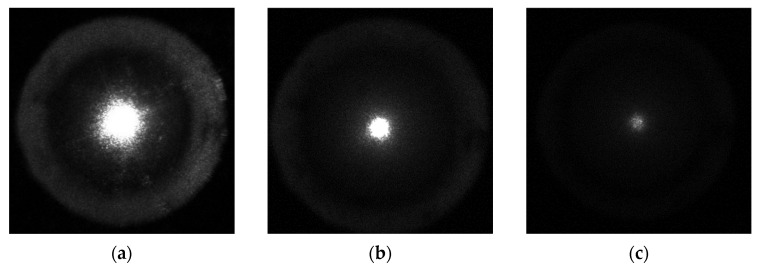
Photometric images obtained by the photometer with ellipsoidal reflectors: (**a**–**c**)—in forward-scattered light; (**d**–**f**)—in backscattered light for different PE film samples’ thickness.

**Figure 4 sensors-23-07700-f004:**
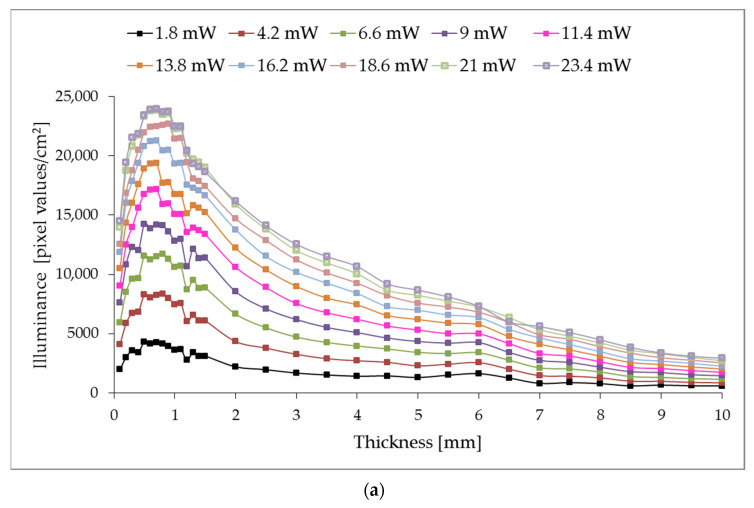
The illuminance of rings A2 (**a**) and A3 (**b**) of photometric images registered by the CCD sensor in transmitted light for different laser beam powers.

**Figure 5 sensors-23-07700-f005:**
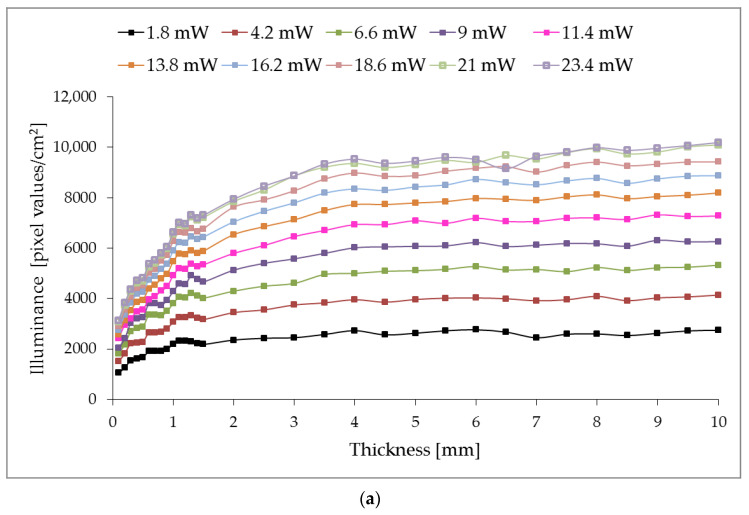
The illuminance of rings A2 (**a**) and A3 (**b**) of photometric images registered by the CCD sensor in reflected light for different laser beam powers.

**Table 1 sensors-23-07700-t001:** Polynomial coefficients in Equations (3)–(6).

Coefficient	Ring A2	Approximation Reliability	Ring A3	Approximation Reliability
*m*	*m*_0_ = 757.62		m_0_ = −127.22	
	*m*_1_ = 1799.9	0.9993	m_1_ = 1761.9	0.9984
	*m*_2_ = −32.413		m_2_ = −25.227	
*n*	*n*_0_ = −0.1335		*n*_0_ = −0.1291	
	*n*_1_ = −0.0521		*n*_1_ = −0.0557	
	*n*_2_ = 0.0105		*n*_2_ = 0.0104	
	*n*_3_ = −0.0011	0.9966	*n*_3_ = −0.0011	0.9991
	*n*_4_ = 0.00006		*n*_4_ = 0.00006	
	*n*_5_ = −0.000002		*n*_5_ = −0.000002	
	*n*_6_ = 0.00000002		*n*_6_ = 0.00000002	
*a*	*a*_0_ = 1397.1		*a*_0_ = −735.69	
	*a*_1_ = 383.68	0.9993	*a*_1_ = 687.74	0.9955
	*a*_2_ = −6.9252		*a*_2_ = −13.634	
*b*	*b*_0_ = 109.24		*b*_0_ = −29.994	
	*b*_1_ = 122.66	0.9983	*b*_1_ = 240.34	0.9957
	*b*_2_ = −2.3069		*b*_2_ = −5.7102	

## Data Availability

Not applicable.

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
