# Peer review of "Optical Properties of Light-Scattering Standards for CCD Photometry"

_sensors, 2023, doi:10.3390/s23187700_

Round 1
Reviewer 1 Report
Dear Authors,
We have carefully reviewed your paper titled "Optical Properties of Light Scattering Standards for CCD Photometry," and we appreciate the valuable contributions your research brings to the field.
Your study showcases the effective use of a photometer with ellipsoidal reflectors as an alternative method for investigating light scattering standards and determining optical coefficients in biomedical experiments. The experimental photometric images obtained using the CCD photometer with ellipsoidal reflectors provided characteristic illumination distributions over the field, enabling a comprehensive investigation of the optical properties of scattering media.
However, the reviewers have raised several questions and concerns that we kindly request you address before finalizing the paper for publication. We believe that addressing these points will strengthen the quality and impact of your work.
1. Could you explain in more detail the principle of operation and advantages of the photometer with ellipsoidal reflectors (ER) compared to other commonly used methods for determining diffuse reflectance and transmission coefficients?
2. How was the 3-D printing process used to fabricate the ellipsoidal reflectors (ER 1 and ER 2) from ABS plastic, and how was the chemical-galvanic metallization carried out on the internal reflective surface? Were there any challenges or limitations in this manufacturing approach?
3. You mentioned the need for pre-calibrating the measuring system with ellipsoidal reflectors using wedge photometry. Could you elaborate on the calibration process and how it ensures accurate measurements?
4. In Figure 2, you showed typical illuminance distributions in photometric images formed in the biomedical photometer with an ellipsoidal reflector. How do you precisely interpret the information from these images and extract the relevant data for determining scattering properties in forward and backward directions?
5. You mentioned the development of specialized software for processing photometric images in reflected and transmitted light, similar to Christian Buil's IRIS. Can you explain the key features and functionalities of this software and how it aids in the automated determination of illumination in rings A2 and A3?
6. Your experiments used different polymers and solutions as biological tissue phantoms to simulate scattering and absorption coefficients. Could you provide more information about the specific properties of these phantoms and how they were prepared to represent the optical properties of biological tissues accurately?
7. The regression analysis resulted in equations (1) and (2) to describe the illuminance of rings A2 and A3 in transmitted and reflected light, respectively. Can you explain the physical significance of the coefficients m, n, a, and b in these equations and how they relate to the laser beam power and the thickness of the examined film samples?
8. You mentioned that the photometric images' illuminance levels depend on the optical properties of the studied samples, such as scattering and absorption coefficients. Could you discuss the potential implications of these findings for applications in structural studies or dynamic monitoring of health indicators?
9. Your study compared the illuminance distribution in photometric images with other spectral measurements of films and coatings of different thicknesses. What similarities or differences did you observe, and how do these findings contribute to understanding illuminance behavior in different optical materials?
10. Towards the end of the section, you mentioned the prospects for further research in search of spectral dependence in the illuminance distribution. What are the potential future directions for this research, and how might it enhance the capabilities and accuracy of the photometer with ellipsoidal reflectors for determining the optical coefficients of biological tissues?
The writer's command of English is evident and devoid of significant issues.
Author Response
- Could you explain in more detail the principle of operation and advantages of the photometer with ellipsoidal reflectors (ER) compared to other commonly used methods for determining diffuse reflectance and transmission coefficients?
Detail the principle of operation and advantages of the photometer with ellipsoidal reflectors (ER) compared to other commonly used methods for determining diffuse reflectance and transmission coefficients are primarily analyzed and presented in previous author's papers, for example [17], [24], [25], [34] in this paper, and another cross references on author's articles. We would not like to increase self-citation to repeat already published rationales in this work.
- How was the 3-D printing process used to fabricate the ellipsoidal reflectors (ER 1 and ER 2) from ABS plastic, and how was the chemical-galvanic metallization carried out on the internal reflective surface? Were there any challenges or limitations in this manufacturing approach?
The authors are grateful for the in-depth analysis of our materials. Still, the details on this issue are already contained in the previously published work of the authors (DOI: https://doi.org/10.20535/1970.53(1).2017.106543), which are freely available. And we would not like to increase self-citation to repeat already published rationales in this work.
- You mentioned the need for pre-calibrating the measuring system with ellipsoidal reflectors using wedge photometry. Could you elaborate on the calibration process and how it ensures accurate measurements?
In the experiments, we used the principles of light attenuation using two polarizing plates and Malus's law principles (this clarification is added to the article text). In our opinion, the principles of such calibration are known, well described, and such that can be classified as general scientific knowledge. Therefore, the authors believe that the addition will contain signs of obviousness.
- In Figure 2, you showed typical illuminance distributions in photometric images formed in the biomedical photometer with an ellipsoidal reflector. How do you precisely interpret the information from these images and extract the relevant data for determining scattering properties in forward and backward directions?
An explanation has been added to the article text – p.5, par.2, following phrases after the first sentence.
“The integral characteristics of diffuse reflection and transmission (total and collimated) on the light scattering standards obtained during reference measurements are formed by the total value of bright pixels across the photometric images entire field. They are correlated with the corresponding values of the diffuse reflectance, total, and collimated transmittance coefficients. These coefficients with the albedo and scattering anisotropy factor [17] are used to determine the media's scattering and absorption coefficients by inverse Monte Carlo simulation [25].”
- You mentioned the development of specialized software for processing photometric images in reflected and transmitted light, similar to Christian Buil's IRIS. Can you explain the key features and functionalities of this software and how it aids in the automated determination of illumination in rings A2 and A3?
The details on this issue are already contained in the previously published work of the authors (https://doi.org/10.21122/2220-9506-2016-7-1-67-76), which are freely available. And we would not like to increase self-citation to repeat already published rationales in this work.
- Your experiments used different polymers and solutions as biological tissue phantoms to simulate scattering and absorption coefficients. Could you provide more information about the specific properties of these phantoms and how they were prepared to represent the optical properties of biological tissues accurately?
We do not show that we correctly understand this remark since the paper is devoted to evaluating the optical properties of specific polymers, which can be used later as standards for determining the coefficients of diffuse reflection and total transmission of biological media.
- The regression analysis resulted in equations (1) and (2) to describe the illuminance of rings A2 and A3 in transmitted and reflected light, respectively. Can you explain the physical significance of the coefficients m, n, a, and b in these equations and how they relate to the laser beam power and the thickness of the examined film samples?
The specified coefficients are a mathematical interpretation of the illuminance changes in different zones of photometric images from thickness for varying incident power levels. At this research stage, the authors are hardly ready to offer the physical context of these coefficients.
- You mentioned that the photometric images' illuminance levels depend on the optical properties of the studied samples, such as scattering and absorption coefficients. Could you discuss the potential implications of these findings for applications in structural studies or dynamic monitoring of health indicators?
This is a significant remark, the answer to which is the global goal of this research topic. The text of the article contains references to the works of the authors [17, 24, 29], in which the potential implications are partially discussed using the example of muscle tissue, liver, and blood. However, larger-scale research requires hardware and ethical methodology refinement, which is definitely in our plans.
- Your study compared the illuminance distribution in photometric images with other spectral measurements of films and coatings of different thicknesses. What similarities or differences did you observe, and how do these findings contribute to understanding illuminance behavior in different optical materials?
- Towards the end of the section, you mentioned the prospects for further research in search of spectral dependence in the illuminance distribution. What are the potential future directions for this research, and how might it enhance the capabilities and accuracy of the photometer with ellipsoidal reflectors for determining the optical coefficients of biological tissues?
The issues raised in questions 9 and 10 are not directly related to the topic of this article. To answer these questions (9 and 10), we can inform you about preparing a new article, which will soon be presented to the public.
Reviewer 2 Report
Denys Bondariev, et al., provided a description of the optical properties of light scattering standards for CCD photometry. The authors made an elaborate effort to describe light scattering standards and how to realize with their proposed method of using a photometer with ellipsoidal reflectors as an alternative method of researching available and cheap light scattering standards. Authors need to make the following changes before considering the manuscript for publication.
1. The explanation of most of the places of the text is very descriptive, and authors need to simplify coming to the point directly.
2. Authors descriptions of most of the experimental data are not clear. The authors mixed the description of some of the applications with the description of the experiments. Please separate and write a separate section for the applications.
3. If possible. authors need to include parameters related to the CCD sensor as well as its role in the overall photoresponse.
The quality of English is OK. But the text is more descriptive and less organized. Authors need to improve in that sense.
Author Response
- The explanation of most of the places of the text is very descriptive, and authors need to simplify coming to the point directly.
- Authors descriptions of most of the experimental data are not clear. The authors mixed the description of some of the applications with the description of the experiments. Please separate and write a separate section for the applications.
(1) and (2) The authors would greatly appreciate more specific suggestions for improving the performance paper. As for the appendices, it does not seem to us that the compilation of this work requires any part to be separated into applications.
- If possible. authors need to include parameters related to the CCD sensor as well as its role in the overall photoresponse.
In our opinion, the aspects mentioned by Reviewer 2 are sufficiently disclosed in the paper text.
Reviewer 3 Report
The manuscript, titled 'Optical Properties of Light Scattering Standards for CCD Photometry', dealt with the calibration process of CCD camera with ellipsoidal reflectors. Experiments were done to verify the designed system with a 650 nm laser.
(1) The title seems not matched with the contents if the readers and the authors consider Photometry and Photometer as different concepts. Obviously the manuscript was testing the optical power of a fixed wavelength laser using CCD cameras. Nevertheless, CCD in the title should also be uppercase (not Ccd).
(2) Figure 1 is not presented in a clear way. What is the BT meaning in the figure? After scrolling up and down several times, I can only guess it is Biological Tissue? Or device under test?
(3) To help reader understand, it is recommended in Fig. 1, there is one sentence to show CCD 1 is for transmission, CCD2 is for reflection.
(4) Should the system consider the exposure time of CCD cameras? Please comment.
Author Response
1) The title seems not matched with the contents if the readers and the authors consider Photometry and Photometer as different concepts. Obviously the manuscript was testing the optical power of a fixed wavelength laser using CCD cameras. Nevertheless, CCD in the title should also be uppercase (not Ccd).
CCD not Ccd - agree
Regarding the title - the authors consider the name correct since Optical properties ... is the result of applying the CCD Photometry method but not the CCD Photometer characteristics.
2) Figure 1 is not presented in a clear way. What is the BT meaning in the figure? After scrolling up and down several times, I can only guess it is Biological Tissue? Or device under test?
Added BT notation the first time the name Biological Tissue is used, in the description of Figure 1 and in its caption.
3) To help reader understand, it is recommended in Fig. 1, there is one sentence to show CCD 1 is for transmission, CCD2 is for reflection.
Has been added to the article text – p.4, second sentence.
“CCD 1 is for detecting transmission and CCD2 – is for reflection light.”
4) Should the system consider the exposure time of CCD cameras? Please comment.
The system allows for programmatically considering the exposure time of CCD cameras; the given results were obtained during measurements with the same exposure.
Reviewer 4 Report
The paper "Optical Properties of Light Scattering Standards for Ccd Photometry" deals with the issue of using CCD sensors in experimental setups to measure the scattering media's optical properties. The background, problem, and aim of this research are clearly presented in the text of this paper. The content is interesting, and the structure and results are understandable. The paper fits the scope of Sensors journal and should be published after "minor revision."
REMARKS:
1. Title: "Ccd" -> "CCD"
2. The primary outcome of this research must be included in the abstract.
3. The introduction is exciting. Is it possible to add a dawning of the described instrumentation setups, e.g., Taylor's method, etc.? It would be perfect for the readers.
4. The technical properties of CCDs used should be included. Is this sensor additionally supported by some spectral filters, e.g., V-lambda?
5. Please explain symbols A1-A3 and R1-R3 in the description of Figure 2. It is described later. But it will make this figure easier to understand.
6. Figures 4 and 5. The unit of illuminance is lux. There is a need to name the measure on the vertical axis in another way.
7. Please indicate the total measurement error of the experimental device in comparison to standard methods and instruments.
8. Your solution proposal is good. Please add information about the total price of your device. Is the device protected by a patent or implemented for sale?
I think that quality of English Language is fine but I am not a native-speaker.
Author Response
- Title: "Ccd" -> "CCD"
Ok
- The primary outcome of this research must be included in the abstract.
Has been added to the abstract.
“Polynomial dependences were obtained, and regression coefficients of the illuminance of the external and middle rings in photometric images for the reflected and transmitted light on the laser power were determined.”
- The introduction is exciting. Is it possible to add a dawning of the described instrumentation setups, e.g., Taylor's method, etc.? It would be perfect for the readers.
Agree with the reviewer. However, adding to the analytical part of the material published at the beginning of the 20th century would be an exaggeration.
- The technical properties of CCDs used should be included. Is this sensor additionally supported by some spectral filters, e.g., V-lambda?
The type of CCD used (a Sony ICX098BL) is indicated in the paper, so the reader can easily find its characteristics if necessary. Additional elements (some spectral filters, e.g., V-lambda) were not used, which is why we did not write anything about it.
- Please explain symbols A1-A3 and R1-R3 in the description of Figure 2. It is described later. But it will make this figure easier to understand.
Has been added to the description of Figure 2.
”R1 is the radius of the collimated flow; R2 and R3 are the radii limiting the peripheral illuminance zone in the photometric image; A1 is the area of a circle with radius R1; A2 is the area of the middle ring bounded by circles R1 and R2; A3 is the area of the external ring bounded by circles R2 and R3.”
- Figures 4 and 5. The unit of illuminance is lux. There is a need to name the measure on the vertical axis in another way.
It is true that the unit of illuminance is lux. The instrument is not calibrated in SI units. Signal presented is in fact illuminance – hence the presentation in other units.
- Please indicate the total measurement error of the experimental device in comparison to standard methods and instruments.
At this stage, the photometer is implemented as an experimental setup, so we consider it correct not to specify the metrological characteristics of the equipment at this time.
- Your solution proposal is good. Please add information about the total price of your device. Is the device protected by a patent or implemented for sale?
At this stage, the photometer is implemented as an experimental setup, so we consider it correct not to specify the metrological characteristics of the equipment at this time. The device is patented in Ukraine.
Round 2
Reviewer 1 Report
The author has made efforts to address some of the queries raised, but there remain specific concerns—namely Questions 7, 9, and 10—that have not been conclusively responded to. I have no further questions or comments to add.
Author Response
- The regression analysis resulted in equations (1) and (2) to describe the illuminance of rings A2 and A3 in transmitted and reflected light, respectively. Can you explain the physical significance of the coefficients m, n, a, and b in these equations and how they relate to the laser beam power and the thickness of the examined film samples?
An explanation has been added to the article text – p.10, par.1, following phrases after the Table 1:
“It is evident that when restoring the illuminance distribution function from the laser power for different thicknesses media samples, the photometric image analysis consists of determining the integral gray level over the image's entire field within the circular region of interest with a radius of R3. At the same time, wedge photometry, as a calibration prerequisite for the functioning of photometry with ellipsoidal reflectors, is aimed at obtaining the range of the linear section of the CCD light intensity response curve. This aspect is fundamental for determining the optical coefficients of the sample under study. When observing the behavior of the coefficients m, n, a, and b in the equations of illuminance of the middle and external ring in transmitted and reflected light, from a physical point of view, it is possible to ascertain the nonlinear dependence of the amount of scattered light on the laser power in different zones of the photometric image. The scattering properties of the selected standards of the corresponding thicknesses can fully explain this. At the same time, the authors predict the existence of a relation between the illuminance and the concentration of scattering centers in the standards, the absolute values of which increase proportionally with the increase in the thickness of the sample.”
- Your study compared the illuminance distribution in photometric images with other spectral measurements of films and coatings of different thicknesses. What similarities or differences did you observe, and how do these findings contribute to understanding illuminance behavior in different optical materials?
We believe that the answer to this question (in the sense available to the authors) is already contained in the text of the article - P. 10, second paragraph.
- Towards the end of the section, you mentioned the prospects for further research in search of spectral dependence in the illuminance distribution. What are the potential future directions for this research, and how might it enhance the capabilities and accuracy of the photometer with ellipsoidal reflectors for determining the optical coefficients of biological tissues?
An explanation has been added to the article text – p.11, par.2:
“Spectral density values characterize the optical properties of biological media. At the same time, photometry using monochrome and color CCD cameras is implemented to analyze illumination by shades of gray. Optimizing the functionality of the photometer with ellipsoidal reflectors and ensuring acceptable accuracy of photometry comes down to the correct selection of the CCD sensor and the range of the used power and spectrum of collimated radiative sources. At the same time, the authors foresee the need to develop a base of reference measurements for different classes of light-scattering media (conditionally more scattering, less scattering, and intermediate values) in the working spectral range of a photometer-specific configuration.”
Reviewer 3 Report
I suggest double check figure captions again
Good
Author Response
In the opinion of the authors, all descriptions of figures and figure captions are complete. If there is a need for correction in any place then please be more specific.